# Indoxyl Sulfate-Induced Macrophage Toxicity and Therapeutic Strategies in Uremic Atherosclerosis

**DOI:** 10.3390/toxins16060254

**Published:** 2024-05-31

**Authors:** Takuya Wakamatsu, Suguru Yamamoto, Shiori Yoshida, Ichiei Narita

**Affiliations:** 1Division of Clinical Nephrology and Rheumatology, Niigata University Graduate School of Medical and Dental Sciences, Niigata 951-8510, Japan; tkywakamatsu@yahoo.co.jp (T.W.); shiori050214@yahoo.co.jp (S.Y.); naritai@med.niigata-u.ac.jp (I.N.); 2Ohgo Clinic, Maebashi 371-0232, Japan

**Keywords:** indoxyl sulfate, macrophage, atherosclerosis, uremic toxins, adsorption, chronic kidney disease, uremia

## Abstract

Cardiovascular disease (CVD) frequently occurs in patients with chronic kidney disease (CKD), particularly those undergoing dialysis. The mechanisms behind this may be related to traditional risk factors and CKD-specific factors that accelerate atherosclerosis and vascular calcification in CKD patients. The accumulation of uremic toxins is a significant factor in CKD-related systemic disorders. Basic research suggests that indoxyl sulfate (IS), a small protein-bound uremic toxin, is associated with macrophage dysfunctions, including increased oxidative stress, exacerbation of chronic inflammation, and abnormalities in lipid metabolism. Strategies to mitigate the toxicity of IS include optimizing gut microbiota, intervening against the abnormality of intracellular signal transduction, and using blood purification therapy with higher efficiency. Further research is needed to examine whether lowering protein-bound uremic toxins through intervention leads to a reduction in CVD in patients with CKD.

## 1. Introduction

Cardiovascular disease (CVD), which encompasses coronary artery disease, peripheral artery disease, cerebrovascular disease, and aortic atherosclerosis, is the primary cause of mortality in patients with end-stage kidney disease (ESKD) [1]. 

The major factors of CVD are atherosclerosis and vascular calcification. Atherosclerosis is accelerated by the malfunction of macrophages, which is caused partially by uremic toxins in cases of chronic kidney disease (CKD) patients, especially those undergoing hemodialysis. In this article, we will review the mechanisms of atherosclerosis progression caused by macrophages and indoxyl sulfate (IS), one of the major uremic toxins.

## 2. Cardiovascular Disease in CKD Patients

While lipid-lowering treatments contribute to the risk reduction of cardiovascular events in the general population [2,3], large randomized controlled trials (RCTs) did not reveal any effectiveness of lipid-lowering therapy in dialysis patients. The Die Deutsche Diabetes Dialyse (4D) trial showed that atorvastatin had no significant effect on cardiovascular death, nonfatal myocardial infarction, and stroke in patients with diabetes receiving hemodialysis [4]. The Assessment of Survival and Cardiovascular Events (AURORA) trial showed that lowering the serum LDL cholesterol levels using rosuvastatin had no significant effect on death from cardiovascular causes, nonfatal myocardial infarction, or nonfatal stroke in hemodialysis patients [5]. The Study of Heart And Renal Protection (SHARP) trial showed that though the reduction in low-density lipoprotein (LDL) by the administration of ezetimibe and simvastatin had a significant effect on atherosclerotic events, including heart attacks, stroke, or operations, to unblock arteries in non-dialysis dependent patients with CKD stages 3–5, the differences were not significant in dialysis patients [6]. These findings suggest that lipid-lowering treatments are insufficient for CKD-induced CVD prevention in ESKD patients. 

The acceleration of CVD in CKD patients could not only be induced by traditional risk factors, such as aging, male, diabetes, smoking, hypertension, lipid metabolism disorder, hyperhomocysteinemia, inflammation, oxidative stress, and family history, but also by CKD-specific factors, such as anemia, clotting disorder, mineral metabolism disorder, sympathetic hyperactivity, and accumulation of uremic toxins [7,8,9]. Previous research about chronic kidney disease–mineral and bone disorder (CKD-MBD) showed that hyperphosphatemia, hypercalcemia, or high PTH levels are associated with CVD events [10]. The Evaluate the New Phosphate Iron-Based Binder Sucroferric Oxyhydroxide in Dialysis Patients with the Goal of Advancing the Practice of EBM (EPISODE) trial, which was an open-label multicenter, randomized controlled trial comparing lanthanum carbonate versus sucroferric oxyhydroxide, or a standard phosphate control (5.0–6.0 mg/dL) versus a strict control (3.5–4.5 mg/dL) in dialysis patients, showed that strict management of serum phosphorus suppressed the coronary artery calcification scores during the 12 months of treatment [11]. In the Dialysis Outcomes and Practice Pattern Study (DOPPS), patients with serum phosphorus levels between 3.6 and 5.0 mg/dL had lower cardiovascular mortality than those managed with levels above 6.0 mg/dL [12]. Also, in the post hoc analysis of the AURORA trial, dialysis patients who were managed with lower serum phosphorus levels exhibited a significantly greater suppression of cardiovascular events and all-cause mortality, which could be attributed to statin use, compared to those with serum phosphorus levels exceeding 5 mg/dL [13]. As mentioned, the management of serum phosphorus is crucial for reducing cardiovascular events in CKD and dialysis patients. With the recent introduction of new phosphate-lowering agents and calcimimetics, achieving the appropriate management of calcium, phosphorus, and PTH has become more feasible. A meta-analysis of 15 large-scale double-blind placebo-controlled trials showed that sodium glucose co-transporter 2 (SGLT2) inhibitors suppress cardiovascular events and improve the prognosis in CKD patients, reducing the risk of cardiovascular death or hospitalization for heart failure [14].

The removal efficiency of uremic toxins, ranging from small-sized molecules to low molecular weight proteins, has improved in the maintenance dialysis patients due to the widespread use of high-performance membranes and hemodiafiltration (HDF) methods. The CONVINCE study reported that high-dose HDF demonstrated a lower risk of overall mortality compared with high-flux hemodialysis [15]. However, the overall removal effectiveness of uremic toxins may still need to be considered because the survival in dialysis patients is still worse than that in the general population [16]. At this point, the increase in serum levels of protein-bound uremic toxins, such as IS and p-cresyl sulfate (PCS), is particularly associated with an increased risk of the progression of renal failure, cardiovascular events, and mortality [17].

## 3. Uremic Toxins and Cardiovascular Disorders in CKD Patients

Accumulating uremic toxins is a major factor in CKD-related systemic disorders. Uremic toxins are defined as follows: (1) Such a compound should be chemically identified, and accurate quantitative analysis in biological fluids should be possible; (2) the total body and plasma levels should be higher in uremic than in nonuremic subjects; (3) high concentrations should be related to specific uremic dysfunctions and/or symptoms that decrease or disappear when the concentration is reduced; (4) biological activity, conforming to clinical changes observed in conjunction with the uremic syndrome, should be proven in vivo, ex vivo, or in vitro; and (5) concentrations in these studies should conform to those found in body fluids or tissue of uremic patients [18,19]. The conventional classification of uremic toxins is based on the report by Vanholder et al. as follows: small water-soluble compounds (non-protein-bound, molecular weight <500 daltons), protein-bound compounds, and middle molecules (>500 daltons) [19]. However, Rosner et al. updated the definition and classification based on their physicochemical characteristics, dialysis removal patterns, and clinical symptoms [20]. The suggested updates are as follows: (1) Solute identification and accurate quantitative analysis in plasma, serum, or blood should be possible; (2) plasma, serum, or blood levels should be higher in CKD than in subjects with normal kidney function; (3) negative effects, conforming with or contributing to biological or clinical changes in CKD, should be proven in vivo, ex vivo, or in vitro, and (4) biologically active concentrations in these studies should conform to those found in plasma, serum, or blood of CKD patients. Among them, small protein-bound molecules (SPBMs) have been shown to exhibit significant toxicity in the arteries and to accelerate atherosclerosis and vascular calcification in basic studies [21,22]. Many SPBMs are produced through the metabolism of dietary proteins by gut microbiota and undergo conjugation reactions in the liver. Due to their binding to essential proteins such as albumin in the bloodstream, SPBMs have low removal efficiency through hemodialysis and tend to accumulate in the body. The accumulation of SPBMs is associated with systemic disorders accompanying CKD, including all-cause mortality, infectious events, cognitive disorders, and pruritus [23,24,25,26,27].

IS, one of the major SPBMs, has a binding rate of 97.7% in the blood of hemodialysis patients, and the reduction rate with conventional dialysis is only 31.8% [26]. Vanholder et al. ranked IS toxicity with the second highest evidence score in SPBMs, such as inflammation, CVD, CKD–MBD, fibrosis metabolic function, and thrombogenicity [28]. IS acts widely across various organs, tissues, and cells such as bones [29,30,31,32,33], skeletal muscles [34,35,36,37,38,39,40,41,42,43,44], and myocardium [45,46,47,48,49,50,51], causing dysfunction in their functions. A clinical study has shown that the serum levels of IS increased with CKD progression, and high levels of serum IS are independently associated with an increased all-cause mortality and cardiovascular mortality in CKD patients [25] and a trend towards increased cardiovascular events in hemodialysis patients [23]. Therefore, understanding the physiological effect of IS on vascular toxicity and the therapeutic strategies for detoxifying IS will be critical for improving mortality, as well as the activity of daily living (ADL) and quality of life (QOL) in CKD patients.

## 4. Uptake of IS into Macrophages

Dysfunction of macrophages within atherosclerotic lesions is a significant feature of atherogenesis. Bone-marrow-derived monocytes in peripheral circulation are recruited into injured vascular endothelium or vascular smooth muscle. These monocytes differentiate into macrophages, proliferating and absorbing oxidized LDL in atherosclerotic lesions [52,53]. Lipid-loaded macrophages, also known as macrophage foam cells, contribute to the development of atherosclerotic lesions by inducing focal inflammation, intimal hyperplasia, or plaque, making it more susceptible to rupture [54,55]. As CKD accelerates the progression of atherosclerosis, we opted to focus on the reaction of macrophages to IS to understand its mechanism and identify therapeutic strategies for uremic atherosclerosis.

With atherosclerotic reactions occurring in the uremic state, IS can lead to the dysfunction of macrophages. IS is a byproduct of tryptophan, an essential aromatic amino acid found in food. After being taken up by small intestinal epithelial cells, tryptophan synthesizes various biological substances, including serotonin, melatonin, niacin, and nicotinamide adenine dinucleotide [56]. A part of the remaining tryptophan is metabolized into indole by gut microbiota in the colon. The gut microbiota composition in ESKD patients significantly differs from healthy controls, as represented by decreases in the Lactobacillaceae and Prevotellaceae families [57]. A previous study revealed that IS levels are controlled by manipulating gut microbiota with diet and genetically modified bacteria [58]. Indole is transported into the liver via the portal vein, where it is converted to IS by hepatic sulfate conjugation. Because the kidney is primarily responsible for removing IS, insufficient elimination of IS in CKD patients leads to increased IS levels in the plasma, which can cause toxicity in various organs and tissues. 

IS has been shown to stimulate monocytes, leading to vascular endothelial inflammation [59]. Exposure of monocytes to IS and p-cresol promotes adhesion, invasion, and migration of monocyte/macrophage through activating the integrin-linked kinase (ILK)/AKT signaling pathway and podosome formation [60]. These biological reactions to IS can lead to the accumulation of macrophages in vascular lesions. 

In humans, organic anion transporters (OATs) and organic anion transporter polypeptides (OATPs) play crucial roles in the uptake of endogenous substances, including IS. OATs transport a variety of molecules, such as cyclic nucleotides, conjugated sex steroids, odorants, uric acid, prostaglandins, and/or metabolites. On the other hand, OATPs primarily transport amphipathic organic anions with a molecular weight of over 300 Da [61]. OATs may be involved in interorgan communication [62]. Organic anion transporter 1 (OAT1) and organic anion transporter 3 (OAT3) are known to be transporters that take up IS in vascular smooth muscle cells, endothelial cells, and proximal tubular cells [63,64,65]. However, there is no evidence of OAT1 or OAT3 expression in macrophages. According to several reports, organic anion transporter polypeptide 2B1 (OATP2B1), a member of the OATP family, may be involved in the uptake of IS into macrophages [66,67]. Knockdown of OATP2B1 in human macrophages reduced the production of pro-inflammatory cytokines [66], and an OATP2B1 inhibitor suppressed IS-induced oxidation and caused a decline in phagocytosis [67]. Therefore, OATP2B1 is the primary transporter of IS in macrophages.

## 5. IS-Induced Macrophage Inflammatory Reaction (Figure 1)

Monocytes circulating in the blood are a heterogeneous population that can be divided into the following three subsets: CD14++CD16− (classical monocytes; cMo), CD14++CD16+ (intermediate monocytes; intMo), and CD14+CD16++ (non-classical monocytes; ncMo) [68]. cMo primarily perform phagocytosis and produce ROS in response to bacterial infections, and under non-inflammatory conditions, they mature into ncMo. ncMo secrete inflammatory cytokines and respond during viral infections. intMo has been implicated in CVD, and an increased proportion of intMo is a predictor of CVD and acute heart failure [69]. In CKD patients, intMo is an independent risk factor for cardiovascular events [70]. The proportion of intMo also increases in chronic dialysis patients, and its extent may serve as a predictor for the complications of CVD [71]. The plasma concentration of IS has been shown to positively correlate with the proportion of CD14+CD16+ monocytes in circulation [72].

Macrophages are classified into two phenotypes based on function, namely the classically activated M1 phenotype and the alternatively activated M2 phenotype. In general, M1 macrophages function in the recruitment of Th1 cells, resistance against microbial pathogens, and regulation of tumor cell metabolism through innate and adaptive immune responses by producing proinflammatory cytokines, such as interleukin-6 (IL-6), monocyte chemoattractant protein-1 (MCP-1), and cyclooxygenase-2 (COX-2) [73]. M2 macrophages function in the clearance of pathogens, anti-inflammatory response, tissue repair, and tumor progression through producing inflammation regulators and tissue-proliferation activators, such as interleukin-10 (IL-10), peroxisome proliferator-activated receptor gamma (PPARγ), transforming growth factor-beta (TGF-β), and tissue inhibitor of metalloproteinases-1 (TIMP-1) [74,75]. Exposure of cells to uremic IS concentrations (100–2000 μM) results in various cellular dysfunctions and the expression of inflammatory cytokines [76,77], gives rise to nuclear factor (erythroid-2-related factor)-2 (Nrf2) downregulation [78], and promotes the differentiation of macrophages towards the M1 phenotype by suppressing delta-like ligand 4 (Dll4) degradation through the inhibition of the ubiquitin-proteasome pathway [66]. The proliferation of M1 macrophages induced by IS is also because of the downregulation of β-catenin, which induces M2 polarization of macrophages [79]. Recently, Klotho, a regulator of oxidative stress and senescence, has been reported to mediate one of the critical molecular mechanisms that may alleviate the toxic inflammatory response caused by IS, by promoting M2 macrophage polarization [80,81]. As kidney function deteriorates, Klotho production decreases [82], which may enhance the IS-induced inflammatory response in macrophages [83]. Another study revealed that a moderate increase in IS promotes monocyte transition into profibrotic macrophages, representing the M2 phenotype [72]. Moderate IS concentrations simulating CKD 1-3 (10 μM or 20 μM) results in the upregulation of AhR and an anti-inflammatory immune response characterized by reduced matrix metalloproteinase-9 (MMP-9) activity and overexpression of PPAR-γ, TIMP-1, TGF-β, IL-10. Additionally, IS stimulation triggers a classical immune response, increasing the production of inflammatory cytokines such as IL-6, C–C motif chemokine ligand 2 (CCL2), and COX-2. Thus, stimulation by moderate IS concentrations induces monocytes to undergo low-inflammatory, profibrotic macrophage polarization, similar to that observed in M2 macrophages [72]. These results suggest that IS primarily induces macrophage differentiation towards the pro-inflammatory M1 phenotype; however, IS may also induce macrophage M2 differentiation. In other words, the composition of M1/M2 phenotype in macrophages may vary depending on the concentration of IS. This dual effect of IS on macrophages may result in the progression of vascular remodeling in atherosclerotic lesions in CKD patients.

IS increases oxidative stress, which plays a crucial role in the development of CVD in CKD patients [84]. IS promotes the production of reactive oxygen species (ROS) partially by stimulating the nicotinamide adenine dinucleotide phosphate (NADPH) oxidase (NOX) pathway in cultured macrophages [67,76,85]. The aryl hydrocarbon receptor (AhR), one of the intracellular receptors of dioxins and dioxin-like compounds, is considered the receptor for IS [86]. AhR stimulation increases p47phox expression, activating NOX and producing ROS [87]. 

Nrf2 is believed to promote the gene expression of phase II detoxifying enzymes and antioxidant enzymes, and the Nrf2–Kelch-like ECH associating protein-1 (Keap1) system is thought to play a role in suppressing oxidative stress and cellular protection [88]. Pedruzzi et al. demonstrated that an impairment of the activation of the Nrf2-Keap1 system could worsen oxidative stress and inflammation in CKD [89]. Furthermore, decreased expression of Nrf2 in peripheral blood mononuclear cells of 20 hemodialysis patients compared to 11 healthy subjects was confirmed [90]. Recently, it was found that IS downregulates Nrf2 expression in HK-2 cells, which is mitigated by NF-κB inhibitors [78]. Therefore, IS, which increases in CKD patients, may contribute to a dysfunction of the Nrf2/Keap1 system. Activation of the antioxidant nuclear factor 2 (Nrf2)–Keap1 pathway, which is known as one of the suppressors of the progression of the inflammatory state [91], may be impaired by IS-induced AhR dysfunction, resulting in the induction of low inflammatory and profibrotic macrophage polarization [92]. This reaction may contribute to maladaptive vascular remodeling and aneurysm formation [72]. IS-induced AhR dysfunction may also partially cause the activation of the mitogen-activated protein kinase (MAPK)/nuclear factor-kappa B (NF-κB) signaling pathway without activating the nucleotide-binding oligomerization domain-like receptor family, pyrin domain-containing 3 (NLRP3) inflammasome [85,93,94,95].

The NLRP3 inflammasome is a significant component that mediates the activation of caspase-1 and secretion of the mature proinflammatory cytokines, interleukin-1beta (IL-1β) and interleukin-18 (IL-18), in response to stimulation by pathogen-associated molecular patterns (PAMPs) or damage-associated molecular patterns (DAMPs) [96,97,98]. Inflammation induced by IS is partially associated with pro-matured-IL-1β proliferation combined with MAPK/NF-κB signaling pathway activation and NLRP3 inflammasome inactivation, resulting in insufficient maturation of IL-1β [93]. This response may cause low-grade inflammation and the promotion of atherosclerosis.

**Figure 1 toxins-16-00254-f001:**
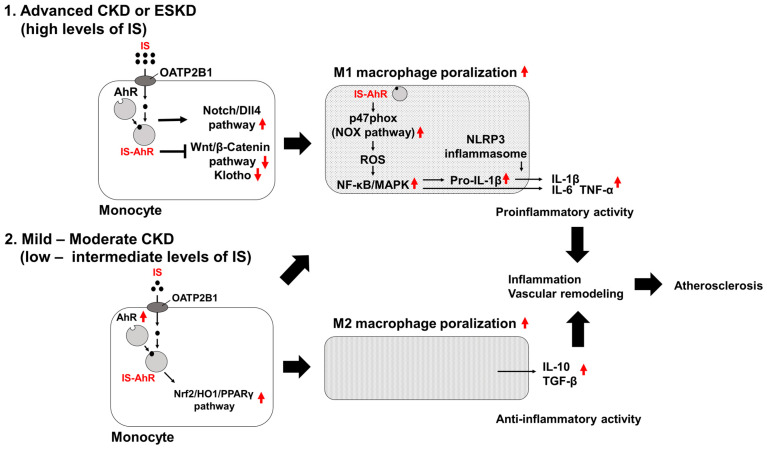
In advanced CKD, high IS levels promote the polarization and proliferation of M1 macrophages. IS is taken up into monocyte via OATP2B1 and binds to the intracellular receptor AhR. This interaction leads to a decrease in the activity of the Wnt/β-catenin pathway and Klotho, while increasing the activity of the Notch/DII4 pathway, inducing polarization towards M1 macrophages. In the M1 macrophages, the reaction between IS and AhR increases ROS production through the NOX pathway, activating NF-κB/MAPK and promoting the production and secretion of inflammatory cytokines (IL-1β, IL-6, TNF-α). However, in this process, the NLRP3 inflammasome, which is responsible for maturing pro-IL-1β into IL-1β, is not activated strongly. On the other hand, in mild to moderate CKD, IS increases the expression of AhR, activating the Nrf2/HO1/PPARγ pathway and promoting M2 macrophage polarization. M2 macrophages primarily enhance the production and secretion of regulators of inflammation (IL-10, TGF-β) and induce anti-inflammatory activity. These reactions may indicate the exacerbation of inflammation, vascular remodeling, and atherosclerosis associated with CKD progression. Abbreviations: AhR, aryl hydrocarbon receptor; CKD, chronic kidney disease; Dll4, Delta-like ligand 4; HO1, heme oxygenase-1; IL-1β, interleukin-1beta; IL-6, interleukin-6; IL-10, interleukin-10; IS, indoxyl sulfate; IS-AhR, indoxyl sulfate-bound aryl hydrocarbon receptor; NF-kB, nuclear factor-kappa B; NLRP3, nucleotide-binding oligomerization domain (NOD)-like receptor containing pyrin domain 3; NOX, NADPH oxidase; Nrf2, Nuclear factor erythroid 2-related factor 2; OATP2B1, organic anion transporter polypeptide 2B1; PPARγ, peroxisome proliferator-activated receptor gamma; Pro-IL-1β, pro-interleukin-1beta; ROS, reactive oxygen species; TGF-β, transforming growth factor-beta; TNF-α, tumor necrosis factor-alpha.

## 6. Malfunction of Lipid Metabolism Associated with Foam Cell Formation Induced by IS in Atherosclerotic Lesions

Impaired lipid metabolism in macrophages is one of the hallmarks of atherosclerosis acceleration, and CKD is a major risk factor in addition to inflammation. Macrophages internalize oxidized low-density lipoprotein (ox-LDL) through scavenger receptor type 1 (SR-A1), type 2 (SR-A2), CD36, and LDL receptor-1 (LOX-1) [99]. Intracellular deposition of cholesterol esters via excessive ox-LDL metabolism results in the formation of foam cells [100]. Foam cells accumulate and undergo the secretion of proinflammatory cytokines and necrosis on the arterial wall, establishing atherosclerotic plaque. THP-1 macrophages stimulated by IS increase CD36 expression and ox-LDL uptake, partly through activation of the MAPK pathway [101]. Ox-LDL is considered one of the inducers of NLRP3 inflammasome activation in macrophages [102,103]. Therefore, in CKD patients, an excessive accumulation of ox-LDL associated with the exposure to IS might enhance macrophage inflammasome activation, leading to increased production of proinflammatory cytokine IL-1β. Serum high-density lipoprotein (HDL) level also decreases with the progression of kidney disease [104,105]. In the early stages of CKD, HDL cholesterol level is associated with increased IS levels [106]. Indole-3-acetic acid positively correlates with the monocyte-to-HDL ratio in CKD patients [107]. Low HDL and non-HDL cholesterol levels are associated with CVD in CKD patients, especially those undergoing dialysis [108,109]. 

HDL is the smallest and densest lipoprotein that contains cholesterols, phospholipids, apoproteins, cholesterol esters, and triglycerides [110]. Normal HDL promotes lipid-anti-oxidation, anti-inflammation, and anti-apoptosis. HDL is biosynthesized when apolipoprotein A1 (ApoA1) acquires cholesterols and phospholipids in the circulation via the ATP-binding cassette transporter [111]. HDL in the uremic state fails these crucial roles of lipid metabolism. A large cohort study in hemodialysis patients showed that all-cause and CVD mortality formed the U curve, signifying the increasing risk with the HDL cholesterol level of between <30 mg/dL and >60 mg/dL [108]. ApoA1, one of the major components of HDL and which stimulates cholesterol efflux through ATP-binding cassette (ABC) transporters, is carbamylated and significantly reduces the extent of cholesterol efflux from macrophages in the uremic state [112]. Macrophages and their related cholesterol efflux are crucial in atherosclerosis formation with HDL dysfunction. HDL isolated from hemodialysis patients showed less cholesterol efflux in macrophages [113]. Uremic HDL also enhances macrophage inflammatory reactions [113]. In addition, IS may induce HDL dysfunction in CKD patients. A cross-sectional study investigating the relationship between IS and HDL cholesterol levels in CKD stages 1–3 showed that increasing IS levels were an independent risk factor of low HDL cholesterol levels. Dyslipidemia in early CKD patients may be associated with IS accumulation [106]. When macrophages are reacted with IS in vitro, uremic macrophages show impaired cholesterol efflux to HDL [85]. IS inhibits the expression of ATP-binding cassette transporter G1 (ABCG1), and the activation of the liver X receptor (LXR) with the LXR agonist, T0901317, improves the reaction [85]. 

Overall, lipid and macrophage dysfunctions and their resulting interactions promote foam cell formation in CKD patients. Notably, IS is a critical factor in promoting lipid-loaded macrophage accumulation. 

## 7. Therapeutic Strategies for Atherosclerosis Caused by SPBMs

### 7.1. Diet and Gut Microbiota

As a dietary therapy for CKD, there is growing attention to the importance of using plant-based ingredients with low-phosphorus protein. In a randomized crossover trial targeting maintenance hemodialysis patients, intervention with the one-week therapeutic diet promptly improved mineral metabolism abnormalities, leading to a decrease in total indoxyl sulfate concentration [114]. Furthermore, in a post hoc analysis of a randomized controlled crossover trial targeting CKD stage 3–4 patients, the therapeutic diet reduced the excretion of acid, IS, and PCS in urine compared to the conventional diet after seven days [115]. These results suggest that a plant-centered diet with reduced phosphorus content may suppress the production of uremic toxins in the body and contribute to mitigating metabolic acidosis and oxidative stress.

The gut microbiota composition changes with kidney disease progression due to uremic toxin production [58,116]. A decrease in the Lactobacillus species was observed in rats with kidney injury, but it was ameliorated by Lactobacillus supplementation [117]. Mishima et al. demonstrated the association between the gut microbiota and the production of uremic toxins through basic research [118,119,120]. These results indicated that the production of food-derived uremic toxins depends substantially on the microbiota. Under germ-free renal failure conditions in mice, IS and other food-derived uremic toxins (PCS, phenyl sulfate, cholate, and hippurate) and short-chain fatty acids were absent in the plasma, urine, and feces [118]. The oral administration of lubiprostone, a ClC-2 chloride channel activator, and the optimized proliferation of bacteria, such as the Lactobacillaceae family or Prevotella genus, decreased IS and suppressed tubular injury, kidney fibrosis, and systemic inflammation [119]. Canagliflozin, a SGLT2 inhibitor, optimized gut microbiota composition via increased cecal short-chain fatty acids, reduced serum IS levels, and suppressed systemic inflammation and kidney injury [120]. SGLT2 inhibitors have also been shown to alter the gut microbiota and potentially reduce the production of uremic toxins derived from amino acids, as revealed in proteomics analyses using mice [121].

According to some animal experiments, the administration of the oral charcoal adsorbent AST-120 contributes to the suppression of bacteria-producing uremic toxin precursors or restoration of the Lactobacillus population and mitigation of systemic inflammation and kidney injury [122,123,124]. Therefore, the reduction in uremic toxins by AST-120 may depend on adsorptive properties and flora-restoration properties. 

In humans, some therapeutic interventions could reduce the accumulation of uremic toxins by optimizing gut microbiota [125]. For example, the intake of a very low protein diet (0.3 g/kg/body weight/day) with keto analogs significantly reduced serum IS levels compared to the intake of a low protein diet (0.6 g/kg body weight/day) by CKD patients not yet on dialysis [126]. Based on various systematic reviews, prebiotics, probiotics, or synbiotics decreased IS and microinflammatory markers without causing adverse effects, partly because of the promotion of Bifidobacterium, Lactobacillus, and Subdoligranulum proliferation [127,128,129,130,131,132]; however, other systematic reviews revealed that these microbiota-derived therapies led to little or no significant change in the circulating IS concentrations [133,134,135]. Although it is controversial whether prebiotics, probiotics, or synbiotics effectively reduce uremic toxins, the effectiveness might be changed by factors such as the duration of use or combination with other drugs.

Curcumin, the primary ingredient of turmeric, is a natural polyphenol with antioxidant and anti-inflammatory effects [136,137]. In an RCT comparing the 12-week curcumin intake group with the placebo group among peritoneal dialysis patients, curcumin tended to decrease plasma PCS concentration, suggesting a potential mitigation of oxidative stress [138]. In another RCT involving hemodialysis patients, curcumin intake was shown to attenuate inflammatory cytokines compared to the control group [139]. While it did not result in differences in plasma IS or IAA levels, it significantly reduced PCS levels [140]. A clinical study observing changes in the gut microbiota following curcumin supplementation in adult patients with CKD stage 3-4 showed a significant increase in Lactobacillus spp. and a tendency of increase in the Prevotella group at 3 to 6 months [141]. Curcumin has been demonstrated in multiple basic and clinical studies to possess anti-inflammatory, anti-fibrotic, and albuminuria-reducing effects, suggesting multifaceted desirable effects even in chronic kidney disease [142].

### 7.2. Inhibition of the Cellular Toxicity of SPBMs

Previous in vitro studies about IS and macrophages revealed that the inhibition of NADPH, NF-κB, MAPK, OATs, OATPs, or AhR exerts anti-inflammatory effects, which might be because of interference with the oxidative stress induced by IS. Resveratrol, which is known to be a polyphenol with properties that attenuate oxidative stress and inflammation, activated the Nrf2 pathway which was downregulated by IS, significantly reduced the malondialdehyde (MDA) and ROS production, and inhibited the IS-induced expression of NF-κB in macrophage-like RAW 264.7 macrophages [143]. 

Resveratrol or other antioxidants may reduce the cellular toxicity of SPBMs, but there are few related studies and insufficient evidence regarding their use. Probenecid, as a potent inhibitor of the OAT, may inhibit the intracellular uptake of IS and suppress the induction of ROS. N-acetylcysteine (NAC), ascorbate (Vitamin C), or alpha-tocopherol (Vitamin E) are well-known antioxidants. Sung CC et al. introduced several results of RCTs regarding the clinical effects of anti-oxidants on clinical outcomes in dialysis patients in a review article [144]. The use of tocopherol did not lead to a significant change in overall mortality among dialysis patients; however, it was reported to reduce cardiovascular disease [145]. Additionally, the use of acetylcysteine reduced cardiovascular events by 30% and stroke by 36% over a two-year observation period in hemodialysis patients [146]. Furthermore, using Vitamin C significantly reduced the level of oxidative DNA products, specifically 8-hydroxy-2′-deoxyguanosine, in the peripheral blood lymphocytes of chronic hemodialysis patients [147]. In basic research, several antioxidants have been reported to reduce oxidative stress caused by uremia. NAC inhibited IS-induced ROS production in NRK-52E cells, a rat-derived renal cell line [148]. Additionally, ascorbate suppressed the IS-induced ROS expression in the vascular endothelial tissues of the rat thoracic aorta [149]. Therefore, antioxidants may alleviate oxidative stress caused by IS and potentially lead to favorable clinical outcomes.

### 7.3. Removal of SPBMs

The choice of appropriate renal replacement therapy (RRT) methods could be crucial for effectively removing uremic solutes. 

There are several negative studies as follows. A systematic analysis of representative uremic toxin removal with hemodialysis (HD), online post-dilution hemodiafiltration (postHDF), and online predilution hemodiafiltration (preHDF) in a single-center crossover prospective observational study showed that the mean protein-bound solutes reduction ratio did not differ between the different treatments, except for PCS with a higher reduction ratio during HDF treatments [150]. Hemodialysis with a medium cut-off dialyzer, capable of removing larger middle molecules (molecular weight 25–60 kDa), was expected to efficiently remove SPBMs but could not reduce IS or other SPBMs [151]. Longer hemodialysis durations, which enable favorable control of the left ventricular mass, blood pressure, or small molecule accumulation, did not significantly reduce plasma IS levels [152]. 

Some studies revealed the differences in the reduction rate of IS or suppression of oxidative stress. Direct hemoperfusion (DHP) with a column containing activated carbon markedly reduced SPBMs, such as IS, PCS, and indole-3-acetic acid (IAA), in the serum samples of HD patients [153]. There are several concerns if we apply DHP in a clinical setting. The column might remove crucial molecules for homeostatic control, or it might be cost-ineffective if we use DHP to improve the prognosis of dialysis patients. Hexadecyl-immobilized cellulose beads (HICB), known as β_2_-microglobulin adsorption column on dialysis-related amyloidosis, also decreased the serum-free IS, IAA, phenyl sulfate (PS), and PCS levels but did not effectively remove SPBMs in dialysis patients [154]. Ultrapure dialysis fluid or vitamin E-coated dialyzers may be adequate to alleviate oxidative stress in the uremic state [155,156]. A post hoc analysis from the previous randomized control trial revealed that high-volume hemodiafiltration could decrease IS and other SPBMs more effectively than high-flux hemodialysis [157]. These methods could indirectly suppress IS toxicity. 

Recently, some research showed that intervention to reduce the albumin-binding properties of IS may be effective for removing IS. Albumin-binding competitors, e.g., ibuprofen, increase the dialytic removal of SPBMs, including IS, in blood [158]. Yamamoto et al. also revealed that the acidic and alkaline pH conditions of human serum weakens the protein-binding affinity of SPBMs, including IS, in vitro [159]. Clinical application of these systems to dialysis fluid circuits may promote SPBMs to inhibit the binding of SPBMs with plasma proteins and effectively remove SBPMs.

In summary, while DHP, HICB, and vitamin E-coated hemodialyzers are considered effective for the further removal of SPBMs, their clinical use is not established due to insufficient consideration of clinical outcomes, cost-effectiveness, and potential adverse effects.

Another strategy for removing IS would be inhibiting adsorption and promoting excretion through the intestine. In a single-center RCT involving maintenance hemodialysis patients, oral administration of activated charcoal for eight weeks reduced the serum urea and phosphate compared to the placebo but did not significantly change the serum IS [160]. AST-120 is an orally administered spherical carbon adsorbent that can adsorb uremic toxins, including the precursors of IS in the intestine. It is used to preserve kidney function and improve the symptoms of uremia in patients with progressive CKD [161]. Oral administration of AST-120 to dialysis patients for two weeks was associated with a 45% reduction in serum IS and a simultaneous significant decrease in oxidative stress markers, such as 8-isoprostane and oxidative albumin. AST-120 decreases necrotic areas, reduces the deposition of IS, and inhibits pro-inflammatory cytokines, such as MCP-1, tumor necrosis factor-alpha (TNF-α), and IL-1β, in the aorta of uremic mice. AST-120 is indicated to stabilize atherosclerotic plaque [162]. Post hoc analysis of the K-STAR study (Kremezin study against renal progression in Korea), which was a prospective, multicenter, randomized control study, with the AST-120 arm (n = 226) and the control arm (n = 239) followed up for 36 months, revealed that AST-120 decreases the risk of cardiovascular events in CKD. Moreover, a decrease in the serum IS concentration due to AST-120 use during the study period of one year was associated with the suppression of CKD progression [163]. Therefore, using AST-120 may be beneficial for preventing CVD progression due to the reduction of IS accumulation in CKD patients [164].

## 8. Conclusions—Future Directions in Uremia Research

We reviewed the aggravation of atherosclerosis caused by the malfunction of macrophages exposed to IS. Based on previous studies, we hypothesized that IS is transported into macrophages through OATP2B1, then binds to AhR and produces ROS, which activates several inflammatory pathways and simultaneously promotes foam cell formation by increasing ox-LDL uptake and impairing cholesterol efflux. Further research is necessary to develop strategies to minimize vascular injury caused by IS in CKD patients. Therapeutic approaches that target the gut microbiota with medication or direct reduction by specialized blood purification may be practical.

## Data Availability

Not applicable.

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
