# Peer review of "Indoxyl Sulfate-Induced Macrophage Toxicity and Therapeutic Strategies in Uremic Atherosclerosis"

_toxins, 2024, doi:10.3390/toxins16060254_

Round 1

Reviewer 1 Report

Comments and Suggestions for Authors

The proposed manuscript “Indoxyl sulfate-induced macrophage toxicity and therapeutic 2 strategies in uremic atherosclerosis” presents a timely and well-balanced overview on this very important topic.

The molecular mechanisms and the clinical background are well-described, the manuscript is easy to read.

I suggest expanding on the following points:

1.  While some effects of IS on macrophages are nicely explained, an overview over changes in expression of molecular components in the monocyte/macrophage lineage in patients with CKD/uremia that might relate to different IS concentrations is missing.

2. The involvement of the Nrf2 system is described, though more detail on Nrf2 activation and activity with progression of CKD to ESKD might be added.

3. Vitamin E-coated dialyzers are discussed, it would be valuable to know, which other antioxidant interventions are evaluated as realistic or promising by the authors in the current context.

Author Response

We would like to thank Reviewer #1 for the careful evaluation of our manuscript. We have added the suggested data and changed the sentences in the manuscript as suggested.

  1. While some effects of IS on macrophages are nicely explained, an overview over changes in expression of molecular components in the monocyte/macrophage lineage in patients with CKD/uremia that might relate to different IS concentrations is missing.

Thank you for your feedback. The fact that monocyte and macrophage differentiation and responses vary depending on the concentration of IS, whether high or low, is an important observation, so we have added this to the text (lines 202-204, and 215-222).

  1. The involvement of the Nrf2 system is described, though more detail on Nrf2 activation and activity with progression of CKD to ESKD might be added.

Thank you for your feedback. The Nrf2-Keap1 system is significantly relevant to IS, and I have added this to the text (lines 235-244).

  1. Vitamin E-coated dialyzers are discussed, it would be valuable to know, which other antioxidant interventions are evaluated as realistic or promising by the authors in the current context.

Thank you for your feedback. There are several studies, including RCTs, regarding the effects of antioxidants on uremia. I have added this to the text (lines 390-406).

Reviewer 2 Report

Comments and Suggestions for Authors

This is an interesting review about IS in CKD and atherosclerosis. Here are some comments which may help to improve the manuscript:

13-15: “The clinical application of these approaches is expected to reduce cardiovascular risk and mortality associated with protein-bound uremic toxins in CKD patients.” This sentence may be toned down as long as no data is available and the many negative outcome studies in CKD/dialysis.

33-47: The authors just focus on the negative outcome studies with lipid-lowering treatments. There were many other negative outcome studies in CKD/dialysis. Moreover, just recently also positive outcome studies (SGLT2, HDF – CONVINCE study) were published; they may also be considered here.

52: should be toxins not toxin

90-91: Check font size

124: Monocytes are a heterogenous cell population consisting of classical, intermediate and non-classical monocytes. Intermediate monocytes are especially proinflammatory and linked to mortality in CKD/dialysis. The authors could also include these aspects into the review.

142: “Although” is not the correct word

145: Is there data how is the effect of IS on the different monocyte subsets?

190: The figure legend should be placed together with the figure. Moreover, the size should be enlarged, and the figure needs to be better explained (e.g. what is the meeting of the small grey boxes)

371: What is meant by “they should purify dialysate”?

372: This is not realistic to use for a standard treatment (e.g. costs, time, effort) – therefore this should be toned down

379: Shortly explain what AST-120 is

Author Response

We would like to thank Reviewer #2 for the careful evaluation of our manuscript. We have added the suggested data and changed the sentences in the manuscript as suggested.

  1. 13-15: “The clinical application of these approaches is expected to reduce cardiovascular risk and mortality associated with protein-bound uremic toxins in CKD patients.” This sentence may be toned down as long as no data is available and the many negative outcome studies in CKD/dialysis.

Thank you for your feedback. As you pointed out, due to the lack of established therapeutic strategies, We have revised the description ‘Further research is needed to examine whether lowering protein-bound uremic toxins through intervention leads to a reduction in CVD in patients with CKD.’ (lines 20-22).

  1. 33-47: The authors just focus on the negative outcome studies with lipid-lowering treatments. There were many other negative outcome studies in CKD/dialysis. Moreover, just recently also positive outcome studies (SGLT2, HDF – CONVINCE study) were published; they may also be considered here.

Thank you for your feedback. We have included recent research findings in the text as well (lines 79-90).

  1. 52: should be toxins not toxin

Thank you for your feedback. We have revised it (line 61).

  1. 90-91: Check font size

 Thank you for your feedback. We have revised it (lines 105-107).

  1. 124: Monocytes are a heterogenous cell population consisting of classical, intermediate and non-classical monocytes. Intermediate monocytes are especially proinflammatory and linked to mortality in CKD/dialysis. The authors could also include these aspects into the review.

 Thank you for your feedback. We have described the changes in monocyte subsets in CKD (lines 181-191).

  1. 142: “Although” is not the correct word

  Thank you for your feedback. I have revised it (line 158).

  1. 145: Is there data how is the effect of IS on the different monocyte subsets?

 Thank you for your feedback. The correlation between IS concentration and changes in monocyte subsets is important, and we made an addition to the text (lines 181-191).

  1. 190: The figure legend should be placed together with the figure. Moreover, the size should be enlarged, and the figure needs to be better explained (e.g. what is the meeting of the small grey boxes)

 Thank you for your feedback. We have moved the legend to the same position as the figure. Additionally, We have separated the figure into M1 macrophages and M2 macrophages, and made the labels for each pathway clearer. I have also adjusted the explanatory text (lines 263-277).

  1. 371: What is meant by “they should purify dialysate”?

 Thank you for your feedback. We have deleted the sentence as it could cause confusion.

  1. 372: This is not realistic to use for a standard treatment (e.g. costs, time, effort) – therefore this should be toned down

 Thank you for your feedback. We have revised the description (lines 443-446).

  1. 379: Shortly explain what AST-120 is

 Thank you for your feedback. We have explained what AST-120 is (lines 450-453).

Round 2

Reviewer 1 Report

Comments and Suggestions for Authors

It was a pleasure to read the manuscript.